# Support Vector Regression Modelling of an Aerobic Granular Sludge in Sequential Batch Reactor

**DOI:** 10.3390/membranes11080554

**Published:** 2021-07-22

**Authors:** Nur Sakinah Ahmad Yasmin, Norhaliza Abdul Wahab, Fatimah Sham Ismail, Mu’azu Jibrin Musa, Mohd Hakim Ab Halim, Aznah Nor Anuar

**Affiliations:** 1School of Electrical Engineering, Faculty of Engineering, Universiti Teknologi Malaysia, Johor Bahru 81310, Malaysia; nursakinahyasmin@gmail.com (N.S.A.Y.); fatimahs@utm.my (F.S.I.); mmjibrin@utm.my (M.J.M.); 2Department of Chemical and Environmental Engineering, Malaysia-Japan International Institute of Technology, Universiti Teknologi Malaysia, Kuala Lumpur 54100, Malaysia; mohdhakim@utm.my (M.H.A.H.); aznah@utm.my (A.N.A.)

**Keywords:** support vector regression, artificial neural network, optimization, particle swarm optimization, genetic algorithm

## Abstract

Support vector regression (SVR) models have been designed to predict the concentration of chemical oxygen demand in sequential batch reactors under high temperatures. The complex internal interaction between the sludge characteristics and their influent were used to develop the models. The prediction becomes harder when dealing with a limited dataset due to the limitation of the experimental works. A radial basis function algorithm with selected kernel parameters of cost and gamma was used to developed SVR models. The kernel parameters were selected by using a grid search method and were further optimized by using particle swarm optimization and genetic algorithm. The SVR models were then compared with an artificial neural network. The prediction results R^2^ were within >90% for all predicted concentration of COD. The results showed the potential of SVR for simulating the complex aerobic granulation process and providing an excellent tool to help predict the behaviour in aerobic granular reactors of wastewater treatment.

## 1. Introduction

### 1.1. Background

One of the well-known wastewater treatment technologies is aerobic granular sludge (AGS), which is able to overcome the limitations of conventional activated sludge (CAS) [1]. AGS has a compact and dense biomass characteristic, and it relies on the growth of granules. Thus, when AGS was inoculated with a bubble column through a sequential batch reactor (SBR), it has been observed to have a short settling time for the formation of AGS [2,3,4]. Since then, AGS technology has attracted the interest of the builders for its advantages, especially in reducing footprints besides increasing the efficiency of the wastewater treatment plant. Aside from that, SBR is a preferred choice for improving AGS technology, because it offers operational flexibility with the reliable and rapid cultivation of AGS.

To facilitate and encourage AGS practical application in the field of wastewater treatment plants, numerous studies from researchers have broadly explored the fundamentals of AGS in SBR [5,6,7,8,9]. To further enhance the system, research on modelling and optimization by using simulation of the data-driven mathematical model would be a great evaluation tool to be used on the system. Furthermore, mathematical model has been proven to be a useful tool to study complex operations of the AGS system [10,11].

The investigation on modelling of the CAS in wastewater treatment plants has become a great tool and well-established practice supported by numerous robust models developed over the years. Despite the extensive application of mathematical models, there is still a limited number of studies, especially in modelling of AGS system of WWTPs.

The main obstacle in the AGS system plant is dealing with complex non-linear biological processes that comprise many internal interactions between the variables and sludge characteristic. The biological processes in AGS are determined by diverse substrates and concentration levels of oxygen. The substrates and oxygen concentration are the results of numerous factors such as granules size, diffusion coefficient, biomass, density, spatial distribution, and conversion rate. All these factors strongly affect each other. Hence, it will be much easier to study the effect of each factor involved through mathematical simulation models compared to experimental setup, which is laborious and time-consuming. Moreover, the combination of a mathematical model with model analysis can provide a robust foundation for the design, operation, and optimization of the biological treatment system [12].

There are two types of models for AGS in WWTPs: mathematical deterministic models and data driven models. Examples of mathematical models found in the literature are activated sludge model no. 1 (ASM1) [13,14], followed by the development of ASM2, ASM2d, and ASM3 models. The ASM2 model extends the ability of ASM1 by involving the biological phosphorus and nitrogen removals [15]. L. Chen et al. introduce another concept of ASM1 and aim at simplifying the existing mathematical model [16]. Regardless of all the availability of ASM models stated, finding the interaction processes and modelling of AGS of WWTPs is still difficult to analyse [16]. More complex systems, which include interaction between biological processes, can be challenging in development and calibration of the differential equations and parameters. Furthermore, expertise knowledge and significant efforts are highly required to calibrate the parameters of ASM models that made the work harder. The calibration must be performed for every stage of treatment system. Therefore, the application of real system AGS of WWTP models can be challenging and cumbersome [17]. Basically, kinetic models are more appropriate for understanding the process on a micro level, i.e., substrate fractions, the rates of change of bacterial groups and by-products. However, it is unsuitable if the goal is for control daily monitoring and prediction.

There is a need to establish a comprehensive and adaptive model that can be used for monitoring daily operation of AGS. Machine learning data-driven models can provide a good alternative from mathematical models in this area. However, there are very limited studies that has been reported on data-driven models of AGS in WWTPs to our knowledge. The employment of data-driven model for AGS is still at its early stages. 

Examples of data-driven modelling that has usually been preferred by most researchers is artificial neural networks (ANN). Originally, they were developed to replicate and understand how the human brain works but rapidly evolved to a wider range of implementation in all fields of applications [18]. The neural network applied interconnected neurons between input, hidden, and output neurons to form a network and model a complex functional relationship [19]. For example, the study from H. Gong et al. [20], trained feed-forward neural network (FFNN) with 205 and 136 datasets for predicting chemical oxygen demand (COD) and removal of total nitrogen (TN), respectively. The performance of the model was compared with multi-layer regression (MLR) model. M.S. Zaghloul et al. [21] trained FFNN with 2600 datasets for modelling biomass characteristics and effluent quality. A modular neural network was used in this study, where the model consists of two sub-models. The prediction output of the first model (i.e., biomass characteristics) was used as the second input of the second prediction model (i.e., effluent quality). On the other hand, M.S. Zaghloul et al. [22] made a comparison between adaptive neuro-fuzzy inference systems (ANFIS) with support vector regression (SVR) with 2920 datasets. Therefore, based on all this research done, the greatness of ANN model in predicting the effluent quality has been proven without a doubt. However, this network faced a problem when it came to predicting a limited number of datasets, i.e., ANN may converge to local minima and face the overfitting problem [23]. 

There are other modelling methods that are increasingly attracting researchers’ interest from time to time in the field of wastewater treatment due to their appealing characteristics and promising generalization performance accuracy, such as SVR. ANN and SVR have similar architectures, but what distinguishes these two types of models is that ANN uses multi-layer connection with weight and activation function to predict non-linear problems, while SVR employs a linear model to implement non-linear class boundaries by mapping input vectors into a high-dimensional feature space for data regression. The principle of ANN follows empirical risk minimization (ERM), which targets minimizing the error of training data, which may converge to local minima due to the gradient descent learning algorithm; therefore, even the well-trained ANN data could face problems such as the overfitting problem. On the other hand, SVR follows structural risk minimization (SRM), meaning that it is easily trapped in local minima and is able to provide multiple solutions [24]. Much research in the field of wastewater has focused on prediction using ANN and SVR, but most of their dataset is larger in size and has not been designed specifically for the purpose of comparative analysis. Hence, appropriate steps in developing a model structure for training ANN and SVR is needed to avoid any bias due to the performance of the prediction model being directly affected by the performance of training model.

One of the well-known characteristics of SVR is the ability to deal with non-linearity relationships. This makes SVR better than neural networks, as it can easily work with non-linear dataset by using the kernel tricks. Apart from these characteristics, some disadvantages restrict the use of SVR on academician purposes as well as industrial platforms. Several parameters need to be defined by the user, which are SVR’s kernel function and hyperparameters. A proper setting of these parameters is needed, as they will affect the generalization performance of the prediction model. This has been the main issue for practitioners who have used SVR. Several recommendations have been given on how to appropriately set SVR kernel parameters, but there is no consensus. The method by V. Cherkassky and Y. Ma [25] presented a practical way to set SVR parameters of cost and gamma. However, this method is based on user expertise, user prior understanding, or through the experimental trial setup. Hence, there is no assurance that the parameter values acquired are truly optimal. In addition, the truly optimal parameter selection is further problematic by the fact that the SVR generalization performance depends on both pairs of SVR kernel parameters together with hyperparameters. This makes it harder to manually fine-tune the hyperparameters, making it essential to use an optimization algorithm method to determine their best value [26].

Hence, the aim of this paper is to investigate the predictive performance of ANN and SVR based on the limited dataset of aerobic granulation in the SBR system under three different temperatures, i.e., 30 °C, 40 °C, and 50 °C, in predicting the behaviour of quality effluent of chemical oxygen demand (COD). The available dataset of aerobic granular sludge is limited due to constraint of experimental works that collect under different high temperatures of SBRs, making it hard to collect. From the final results of the ANN and SVR predicting model, that of SVR still needs a process of optimization. Hence, further optimization will also be implemented in this work.

The major contributions of this work can be summarized as:To the very best authors’ knowledge, this is the first modelling work applying on limited dataset of 21 for aerobic sludge granulation using sequential batch reactor at high temperatures, which are 30 °C, 40 °C, and 50 °C.As a comparison, SVR and ANN models were developed to identify the best predictive model on limited dataset, which contains only 21 samples.Instead of using a trial-and-error method, an improvement has been made by using grid search algorithm in determine suitable pairs of *C* and *γ* value.Improvement on conventional grid search algorithm is the proposed by using optimization method using particle swarm optimization (PSO) and genetic algorithm (GA) to give better estimation of the best pair of *C* and *γ* value. The proposed method shows an improvement in accuracy performance of the COD concentration with a limited dataset.

The models for both SVR and ANN are developed and validate and evaluate well in this study to ensure the best models are built.

### 1.2. Optimization Methods

In this work, a proper setting is needed to simulate the SVR model. Out of several methods used from other applications in tuning the parameters, most are using trial-and-error or grid search method. When applying the grid search method, there is a need to step down or decrease the parameter range to enhance the accuracy of the truly optimal solution. However, this method will cause time-wasting for the searching process. Parameter selection can be regarded as a constraint of SVR. Some researchers came out with an idea to solve this problem. Authors of [27] proposed an approach for tuning the SVR parameters by using mutative scale chaos optimization. Meanwhile, [28] used a novel optimization based on the quantum-inspired evolutionary algorithm method.

There are several optimization methods that were developed as fast alternative solutions to replace the grid search method, which are Particle Swarm Optimization (PSO) and Genetic Algorithm (GA). Many works from the literature show that the SVR model that is optimized by methods shows a significant improvement in terms of accuracy performance [29,30]. The motivation for selecting the PSO as an optimization method is due to its speed and simple adjustability of the algorithm [31]. A comparison needs to be made to show the effectiveness of both methods. GA was chosen as another optimization method for determining the parameters of SVR. Both PSO and GA methods have been thoroughly qualified and have been tested under the same conditions. Furthermore, it can be easily implemented using MATLAB software, make it easily parallelized [31]. In addition, both optimized PSO and GA are well-known with their efficiency to locate the global optima, even when the objective fitness function is discontinuous [32]. Hence, it would be interesting to make a comparison in the context of SVR parameters between the optimization method of PSO, GA, and grid search method for evaluating their performance achievements.

## 2. Materials and Methods

### 2.1. Study Area

The data collection of wastewater treatment plants has been conducted in Madinah, Saudi Arabia. During summertime, the temperatures of Madinah climate could reach close to 50 °C. The influent of wastewater is mainly treated in three stages: mechanical, biological, and chemical treatments. During mechanical treatment, all the matters pass through grit removal and fine screening processes. All the small raw will be suspended in the water, while larger raw plastic materials will be screened and ground before they can be conveyed to the next stage. This is to avoid blocking the valve nozzle and pipeline and to prevent damage of the sludge removal equipment.

The biological treatment of this study is based on aerobic granular sludge (AGS) instead of conventional activated sludge systems. The collected sludge was used to cultivate aerobic granulation using a sequencing batch reactor (SBR). Generally, the experiments have been conducted in a double-walled cylindrical glass column bioreactor. The capacity volume of the bioreactors is 3 L with internal radius of 3.25 cm and total height of 100 cm [33]. The system pump including the inlet, outlet, and aerator were all controlled by a programmable logic controller (PLC). The operation of the bioreactor was operated under SBR at a cycle of 180 min: 60 min of inlet feeding pump without mixing, 110 min for aeration, and 5 min each for settling and effluent discharge. Diffusers were placed at the bottom of volumetric flow rate of 4 Lmin^−1^ during aeration. The effluent was discharged through the outlet port located in the middle of the glass bioreactor column with high volumetric ratio of 50%. By heating only half of the column, it was sufficient to achieve the required temperature for the experiment (to develop the granules and process the wastewater). The heat is transferred and distributed evenly inside the bioreactor through conduction and convection. Heat is produced by the heating mantle, and through conduction, heat is transferred from the mantle to the glass column. In addition, heat transfer by convection is driven by the movement of water/fluids during the aeration process. Three sequencing batch reactors (SBR) with different temperatures of 30, 40, and 50 ± 1 °C were controlled using water bath sleeves and thermostats namely as SBR30 °C, SBR40 °C, SBR50 °C, were used for data collection. Figure 1 shows the schematic diagram of AGS experimental laboratory pilot-plant system.

### 2.2. Characteristic and Statistics Analysis of Wastewater Dataset

In this case study, the available dataset obtained for modelling and optimization is limited due to the limitation of experimental works in [33]. The composition that was fed into the rectors is the same composition of the influent used by [34]. The composition of the synthetic wastewater was involved two solutions (A) CH_3_COONa 65 mM, MgSO_4_.7H20 3.7 mM, KCl 4.8 mM, and (B) NH_4_Cl 35.2 mM, K_2_HPO_4_ 4.4 mM, KH_2_PO_4_ 2.2 mM and trace element solution 10 mL^−1^. These solutions were mixed and prepared with distilled water prior to feeding. The parameters considered for this study were total nitrogen (TN), total phosphorus (TP), ammonia nitrogen (AN), total organic carbon (TOC), mixed liquor suspended solids (MLSS), and chemical oxygen demand (COD) as shown in Appendix A. Only effluent COD is considered as the output for model prediction. The overall data collected for 60 days with an interval of 3 days is 21 samples. The experimental work in [33] provides the influent and effluent parameters used for model development. The experiments were conducted in three operating conditions (30, 40, and 50 °C) for verification. Table 1 refers to statistical analysis results of the influent and effluent parameters based on the available dataset obtained from the experimental works in [33].

### 2.3. Data Pre-Processing

The important flow of modelling is data pre-processing. Data pre-processing includes data normalization and data division. In this study, the data has been set to an appropriate range for network learning in the model development. The following steps of modelling are applied to collected dataset of AGS.

Data normalization is performed for the development both models. This is a crucial step in modelling to prevent the domination of a large numeric scale from a smaller numeric scale. It also helps to simplify the difficulties during calculation. This process minimizes the chances of underflow and overflow of data. Normalization data is recommended to scale the training and testing data within the range of (0, 1). Equation (1) is used to normalize the dataset as follows:(1)$${x^{\prime }}_{i}=\frac{x_{i}}{x_{imax}}$$
where $x_{i\;}$ is the input/output data, $x_{imax}$ is the maximum input/output data, and ${x^{\prime }}_{i}$ is the minimum input/output data.

Data division of pre-processing is where the experimental data are divided into training and testing datasets. As stated by P. Samui and B. Dixon, no specific rule is needed to determine the division of training and testing datasets [35]. However, there are several methods recommended that can be applied depending on the size of the experimental dataset for instant *K*−1 fold cross-validation (CV) [36] and hold-out method [37]. In this study, *K*−1 fold CV method was chosen to split the dataset. The CV resampling method is a more suitable one to evaluate the data, especially for the limited dataset case. The single parameter *K* refers to several groups which experimental data is going to split into. In this work, a 10-fold CV is chosen, as it is commonly used and suggested [38]. The *K*−1 fold will be used to train the dataset, while the last fold, as a test dataset. From the iteration of the fold, an average model is then calculated and finalized. Then, the average training model is tested again on the testing dataset. Figure 2 describes the *K*−1 fold CV method.

### 2.4. Artificial Neural Network (ANN)

Artificial neural network was first established to mimic the work of the human brain but rapidly progressed to a broader range of application [18]. It is quite popular, especially for simulating complex processes such as wastewater treatment. The neural network structure consists of three main perceptron layers: input layer, hidden layer, and output layer. Each layer contains neurons and is connected to the next layer of neurons via synapse connections that carry weights. Each neuron will perform mathematical operations on its inputs, and the weight in the neurons will represent the relative significance of the node output. The output layer collects all the data inside the network and transforms it into the desired output, where every output has its own node [39].

#### ANN Structure

The determination of the ANN structure is important and requires additional effort to accomplish. The network structure of ANN is selected by considering the relationship proposed by L. Rogers and U. Dowla [40]. Meanwhile, the range of hidden neurons for the simulation started from 1 hidden neuron up to 16, with an increment of 2, as proposed by R. Hecht-Nielsen [41]. The upper bound is set as the number of hidden neurons to guarantee the good accuracy performance of the model. 

There are two types of processing functions inside the neuron, which are propagation and activation function. In neurons, Equations (2)–(4) is shown as follows [42]. Figure 3 depicts the single neuron architecture of ANN models, which contain an influent input of 6 layers, hidden layer (up to 16 layers), and 1 predicted output layer.

Propagation function:(2)$$v_{i}=\overset{m}{\underset{j=1}{\sum }}w_{ij}x_{i}\left(t\right)-\theta_{i}$$Activation function:(3)$$\sigma \left(f\right)=\{\begin{array}{cc} 1, & \;\;f\geq 0\\ 0, & \;\;Otherwise \end{array}$$Node output:
(4)$$y_{i}\left(t+1\right)=\sigma \left(\overset{m}{\underset{j=1}{\sum }}w_{ij}x_{i}\left(t\right)-\theta_{i}\right)$$ where input is denoted as $x_{i}$, weight as $w_{ij}$, bias is $\theta_{i}$, and time step is t.

The most preferred algorithms are feed-forward neural networks (FFNN), which was used in this work [18]. The simplest design to describe a single-layer FFNN is by using the matrix form. Inputs $x_{i}$ are multiplied by weights $w_{ij}$ to compute the output of the propagation function, which is activation potential $v_{i}$. Equations (5) and (6) demonstrate the activation potential that goes through the activation function σ to obtain the output $y_{i}$.
(5)$$\left[\begin{array}{c} v_{1}\\ v_{2}\\ \vdots \\ v_{i} \end{array}\right]=\left[\begin{array}{ccc} w_{1,1} & \cdots & w_{1,j}\\ w_{2,1} & \cdots & w_{2,j}\\ \vdots & \cdots & \vdots \\ w_{i,1} & \cdots & w_{i,j} \end{array}\right]\left[\begin{array}{c} x_{1}\\ \vdots \\ x_{2} \end{array}\right]-\left[\begin{array}{c} \theta_{1}\\ \vdots \\ \theta_{2} \end{array}\right]$$
(6)$$\left[\begin{array}{c} y_{1}\\ y_{2}\\ \vdots \\ y_{i} \end{array}\right]=\left[\begin{array}{c} \sigma (v_{1})\\ \sigma (v_{2})\\ \vdots \\ \sigma \left(v_{i}\right) \end{array}\right]$$

The key to accomplish the highest precision is to find the right combination of weights, which is accomplished through learning algorithms of neural networks. This is known as training model, where the learning process is started from the pairing of input and output data and continued to improve the weights until the output error is saturated [43].

### 2.5. Support Vector Regression (SVR)

The roots of Support Vector Regression (SVR) was initially introduced by V. Vapnik et al. based on the concept of structural risk minimization [44]. Generally, SVR is a technique that creates a predictive model that is capable in classifying unknown patterns into their groups. Then, SVR application introduced the regression and estimation function [45,46]. Theoretically, SVR depends on a decision planes that separate two different classes in term of margin maximization. It creates a hyper lane by using a linear model to implement non-linear class boundaries through some non-linear mapping input vectors into a high-dimensional feature space. There is non-linear and unknown dependency in mapping function y=f(x) between high-dimensional input vector x and scalar output y. There is no information underlying joint probability functions, and one must contribute a distribution-free learning. Hence, the main goal of SVR model is to create a margin between these two categories as large as possible.

To build a strong predictive model of SVR regression, different mapping kernel functions can be selected into the algorithm. The input dataset is denoted as $\left\{\left(x_{i},y_{i}\right),i=1,2,3,\ldots i\right\},$ where $x_{i}$ is the input and i stands for numbers of data, and it is same as the size of the input dataset. Meanwhile, $y_{i}$ is corresponding to the output data. Despite efficient utilization of high-dimensional feature space of SVR, several other merits of SVR are a distinctively solvable optimization problem, having a theoretical analysis ability using computational learning theory, and having a highly effective approach to build a mathematical model with limited training datasets [47,48].

The basic theory underlying behind SVR algorithm is expressed as follows in Equation (7) [49]:(7)y=f(x)=(ω.ϕ(x))+b
where ω is denoted as the weight vector, ϕ(x) as a non-linear mapping function from low to high-dimensional in a feature space and b is designated as a bias. The value of b and ω in Equation (8) can be derived by substituting the input dataset of $x_{i},y_{i}$ into the Equation (7):(8)$$R_{reg}\left[f\right]=R_{emp}\left[f\right]+\lambda {∥\omega ∥}^{2}=\overset{S}{\underset{i=1}{\sum }}C\left(e_{i}\right)+\lambda {∥\omega ∥}^{2}$$
(9)$$e_{i}=f\left(x_{i}\right)-y_{i}={\hat{y}}_{i}-y_{i}$$
where C is a cost parameter that acts as a regularization parameter by controlling the loss of input dataset; meanwhile, λ is the compromise of model complexity. $R_{reg}\left[f\right]$ represents the sum of experience risk, and $R_{emp}\left[f\right]$ is the empirical risk. A poor choice of C will lead to poor prediction of the SVR model.

A margin classifier is defined for a minimum distance between any input data points and its hyper lane. To accomplish the minimal goal of structural risk, which is the optimal hyper lane, larger margins should be obtained to get better generalization. The higher or smaller confidence risk is denoted as ${∥\omega ∥}^{2}$, and it also reflects the complexity of the model; C(.) is the size of training dataset and loss function, respectively, and $e_{i}$ is the difference between the predicted data ${\hat{y}}_{i}$ and experimental dataset $y_{i}.$ Based on the SVR principle to structured risk minimization, the SVR model sought to minimize the sum of confidence risk and empirical risk. 

The problem in finding function f and given a loss function can be solved using a quadratic programming, as following in Equation (10):(10)$$\max J=-\frac{1}{2}\overset{s}{\underset{i,j=1}{\sum }}\left(\alpha_{i}-\alpha_{i}^{*}\right)(\alpha_{j}^{*}-\alpha_{j})\left(\phi \left(X_{i}\right),\phi \left(X_{j}\right)\right)+\overset{s}{\underset{i=1}{\sum }}\alpha_{i}^{*}\left(Y_{i}-\varepsilon \right)-\overset{s}{\underset{i=1}{\sum }}\alpha_{i}^{*}\left(Y_{i}\right)$$
$$s.t.\{\begin{array}{c} \overset{s}{\underset{i=1}{\sum }}\alpha_{i}\\ 0\leq \\ 0\leq \end{array}\begin{array}{c} \overset{s}{\underset{i=1}{\sum }}\alpha_{i}^{*}\\ \alpha_{i}\leq C\\ \alpha_{i}^{*}\leq C \end{array}\}$$

The value of b and $\sum_{i=1}^{s}\left(\alpha_{i}-\alpha_{i}^{*}\right)(\phi \left(x_{i}\right))$ in Equation (11) can be obtained by substituting any supported number vector into Equation (10):(11)$$f\left(x\right)=\overset{s}{\underset{i=1}{\sum }}\left(\alpha_{i}-\alpha_{i}^{*}\right)\left(\phi \left(x_{i}\right),\phi \left(x\right)\right)+b$$

Several kernel functions are available in literature. To define the kernel function in the inner product of high-dimensional feature space of SVR is as following Equation (12):(12)$$K\left(x_{i},x_{j}\right)=\left(\phi \left(x_{i}\right),\phi \left(x_{j}\right)\right)$$

By computing the kernel function in low-dimensional space, the inner product in high-dimensional space can be obtained. Finally, Lagrange multiplier and optimal constraints are introduced, and the decision function has the following explicit form:(13)$$f\left(x\right)=\overset{s}{\underset{i=1}{\sum }}\left(\alpha_{i}-\alpha_{i}^{*}\right)K\left(x_{i},x\right)+b$$

#### 2.5.1. Kernel Functions

Kernels or kernel functions are a set of mathematical SVR algorithms that play an important role in SVR regression and its performance. The wrong selection of kernel functions will result in accuracy and prediction of the model. Basically, the kernel’s function takes input data and transforms it into the required form. Kernel functions enable the operations to be performed in the input data space rather than in high-dimensional feature space. Different kernels use different types of SVR algorithm. In this study, radial basis function (RBF) (Equation (14)) is used for developing the SVR model.
(14)$$K\left(x,y\right)=\exp(-{∥x-y∥}^{2})/\left(\delta^{2}\right)$$
where y regulates the effect of prediction variations on the kernel variation.

In general, RBF is the most used type of kernel by researchers. RBF kernels non-linearly map sample data into a higher dimensional feature space, unlike linear kernel functions, which are suitable for cases when the relation between the input and output is non-linear. In addition, sigmoid kernels behave like RBF, but only for certain parameters [50]. Polynomial kernel has more parameters than RBF; the number of hypermeters will result in the complexity of model selection. Compared to RBF kernel, it is localized and has a finite response along the entire axis. However, the choice of kernels is different based on application problems, scaling methods, and parameters. Figure 4 depicts the SVR model structure.

#### 2.5.2. Cost and Gamma Parameters

There are two parameters that are considered for RBF kernel function, which are cost of penalty C and gamma γ value. The parameter C will control the trade-off between the slack variable size and margin, and parameter γ influences the partitioning outcome in the feature space of RBF kernel. The selection of suitable C and γ values is important, as it will affect the performance prediction of the model. In determining the parameters, there are several steps: trial-and-error implementation, grid search, or using an optimization method such as a genetic algorithm (GA). Grid search is a conventional way, and it is time-consuming [51]. Alternative ways to find the values of the parameter is through particle swarm optimization (PSO) and genetic algorithm (GA).

### 2.6. Data De-Normalization

Once the prediction output is obtained, the final step of ANN and SVR model is data de-normalization. The purpose is to reverse the normalization process. It is essential to reshape the data into its original scale for the evaluation performance purpose. Figure 5 shows the modelling flow chart of ANN and SVR.

### 2.7. SVR–PSO Prediction Model

In SVR–PSO, the kernel selection parameters and regularization can be obtained through optimization framework that is derived based on particle swarm optimisation method. It based on the population search method that makes use of the idea of shared information by the community. The particles in the PSO are flown through the hyperdimensional search region. The position changes based on social psychology tendency, followed by the achievement of other particles. Therefore, the change of a particle within the dimensional space is inspired by the skills and information acquired by its neighbours. Hence, the results of particles will successfully return towards previously successful regions in hyperdimensional space. 

As mentioned earlier in Section 1, to implement the SVR–PSO approach, kernel parameters (C  and γ) need to be optimized. The position between particles within the swarm is denoted as a vector. It will encode the parameters in SVR. The objective function or fitness function will set up following regression accuracy performance where MSE and R^2^ are used. Thus, the particles in the swarm with high regression accuracy will produce the value of MSE closer to 0 or closer to 1 for R^2^. Figure 6 shows the flow of PSO optimization.

### 2.8. SVR–GA Prediction Model

The concept of GA is intended to mimic natural systems processes, especially for evolution. The GA algorithm framework is based on the principle of the survival of the fittest member in a group of the population that tries to preserve genetic information from one generation to another generation. One of GA’s features is its inherent parallelism to another algorithm that is serial. If one path turns out to be a dead end, it will automatically be eliminated and resumed by other promising paths. Moreover, even on the condition of complex problems, GA is still able to find global optimum among many local optima. This is one of the remarkable capabilities of GA. 

Initially for SVR–GA, the process begins to generate an initial random population. Then, fitness values for each chromosome will be evaluated. The new population will be generated using selection and crossover. Selection is where a new generation is produced. The most fit generation will be nominated, while the least fit will be eliminated. Through the crossover of DNA strands that occur in the reproduction of a biological organism, the new generation will be created to represent the current population. Lastly, when the stopping condition is achieved, the best values are obtained. Otherwise, it would start again with the selection. The stopping condition is satisfied when the lowest MSE values are achieved. Figure 7 shows the flow of GA optimization.

### 2.9. Model Validation

The validation and evaluation of a model involves analysing the regression’s best fit. The plotted graph is checked through several criteria, which reveals which residual plots are random and checks whether the performance of prediction models deteriorates substantially when unknown data (testing data) are applied. The results are compared based on the residual sum of square due to error, RSS (Equation (15)); correlation coefficient, R^2^ (Equation (18)); root mean square error, RMSE (Equation (19)); and mean square error, MSE (Equation (20)).
(15)$$RSS=\overset{N}{\underset{i=1}{\sum }}w_{i}{\left(y_{i}-{\hat{y}}_{i}\right)}^{2}$$
where the number of predictions is defined as N, experimental data as $y_{i}$, $\bar{y}$ as the mean of the predicted data, and ${\hat{y}}_{i}$ as predicted data. The statistic measures the total deviation of the forecast values from the fit to the forecast values. The closer the value is to zero indicates the model has a smaller random error component and that the fit is good for prediction.

The correlation coefficient *R*^2^ has been defined as the ratio of the sum of squares of the regression (*SSR*) and the total sum of squares (*SST*). The closer the value of *R^2^* is to 1 shows that a greater percentage of variance is accounted by the developed model.
(16)$$SSR=\overset{N}{\underset{i=1}{\sum }}w_{i}{\left({\hat{y}}_{i}-\bar{y}\right)}^{2}$$
(17)$$SST=\overset{n}{\underset{i=1}{\sum }}w_{i}{\left(y_{i}-\bar{y}\right)}^{2}$$
where *SST* = *SSR* + *RSS*. Equation (11) expressed the *R*^2^ as:(18)$$R^{2}=\frac{SSR}{SST}=1-\frac{RSS}{SST}$$

Root mean square error, *RMSE*, is also known as the fit standard error and the standard error for the regression problem. A value closer to zero indicates the best fit that is useful for prediction.
(19)$$RMSE=s=\sqrt{MSE}$$
where *MSE* is the square error of the residual mean square.
(20)$$MSE=\frac{RSS}{v}$$

## 3. Results and Discussion

### 3.1. SVR Training

The accuracy performance of SVR model is shown in Table 2. To obtain these results, the hyperparameters C and γ needed to be optimized. The hyperparameters in this study were tuned using the grid search method, which will yield the best value of kernel function cost C and gamma γ, paired with the highest performance. The accuracy performance was measured in terms of RMSE performance. The tuning hyperparameters using the grid search method will control the penalty due to a large prediction of residual errors. The value of C and γ was varied between $\{2^{-15},2^{-14},\ldots ,2^{14},2^{15}\}$ on a log scale function. The tuning using the grid search indicates a high risk of overfitting, hence cross-validation (CV) using K−1 fold with unseen data was essential to ensure the models are generalizable. 

The combinations of C and γ are performed on the training dataset by using the grid search method with 10-fold CV. The 9-fold of 53% of training will undergo the process of searching parameters and leave 1 last fold to be used as a test dataset. The final average model of the training dataset is then calculated before it will be tested again on the remaining fold of 47% testing dataset to verify and validate the results. The best pair of C and γ will provide the best CV accuracy. The epsilon *ε* is fixed to 0.01. The performance of the CV is evaluated based on the residual sum of square (RSS) and RMSE value. The RSS assesses the mean of dependent variables, which means smaller RSS represents better accuracy for the model. From the pair of C and γ in $\{2^{-15},2^{-14},\ldots ,2^{14},2^{15}\}$, the grid search produces 31 × 31 = 961 combinations.

The kernel scale of hyperparameters controls the kernel function response towards the variation in predictions. Low kernel scales imply a milder kernel drop, and large kernel scale imply that the kernel will rapidly drop. Hence, kernel scale is related to spread of the data. Smaller values of kernel scales produce more flexible models.

### 3.2. ANN Training

The choice of network structures greatly effects the performance of ANN models. ANN consists of three layers of neurons, which are used for predicting the relationship between six influent inputs of aerobic granular sludge with the output of COD concentration. In this ANN training, the data were normalized before being separated into 53% training and 47% testing datasets. The range of hidden neurons was varied from 1 hidden neuron with increment of 2 up to 16 hidden neurons. Bayesian regularization and feed-forward neural network (FFNN) algorithms are applied to the training model. The developed FFNN models of COD concentration were analysed by computing R^2^, MSE, and RMSE accuracy performance. Once the prediction output was obtained, the data was de-normalized again to scale back to its original scale. The selected hidden neurons are chosen based on the highest correlation achieved in testing results, as shown in Table 3.

### 3.3. Model Evaluation between ANN and SVM

The SVR and ANN models were evaluated using a very limited dataset of 21 under different high temperatures. The testing dataset were isolated from the beginning during the development of training models to ensure adequate evaluation. Figure 8 shows the comparison of plotted graph between SVR and ANN models during training and testing for both models at three different temperatures. 

As shown in Figure 8, the training of the SVR model was able to predict the COD concentration for SBR30 °C, SBR40 °C, and SBR50 °C with R^2^ values of 98.41%, 96.69%, and 96.17%, respectively. The testing of the SVR model’s performance resulted in 89.29%, 88.43%, and 93.43%, respectively. Meanwhile, the training for the ANN model achieved 72.42%, 96.06%, and 92%, and for testing 82.62%, 85.39%, and 89.10%, respectively. The main aim of three different high temperatures used in this study was to investigate the granulation process, density, and performance stability of AGS in treating the wastewater.

From the results, the SVR model showed a clear advantage over FFNN model in term of R^2^ and MSE. This is because the available amount of dataset of AGS in this study is limited, which can cause the training of the neural network to be disrupted by noise, especially in wastewater treatment research. Noise can occur due to input used and measurement errors [52,53]. Hence, it proved that SVR is a better option in modelling a limited and complex dataset such as AGS.

The prediction performance of ANN is strongly influenced by the number of neurons in the hidden layer, from 1 to 16, that have been trained and tested with random combinations of weights and threshold values. This makes the ANN model unstable due to changing network architecture, as the network was changing every time it was trained and tested. Besides, it is noted that ANN with a smaller number of neurons will have more instability, especially networks with one or two neurons in the hidden layer.

The dependent of the SVR model towards the training dataset is also less as compared to the FFNN model that required a larger dataset to obtain the desired accuracy performance. In fact, the SVR model is based on statistical theory, and it has rigorous mathematical and theoretical foundation, while the FFNN model needs to rely on the designer’s knowledge and experience [54]. Hence, it could be concluded that the SVR model approach can give higher accuracy compared to FFNN using limited dataset in predicting COD concentration. Table 4 shows the performance comparison between SVR and FFNN in terms of R^2^ and MSE for different temperatures of reactors.

The studies of modelling of aerobic granulation are still in their infancy. Three previous studies of aerobic granulation were performed by [21,22,55]. Table 4 tabulates all the outcomes of these models, including this work. The ANN model was developed by H. Gong et al. [20] to investigate the quality effluent of total nitrogen (TN) and COD. The collected dataset used was 205 and 136 samples. The model used single hidden layer; unfortunately, the number of neurons was not reported. The final performance is usually based on prediction of the unseen dataset, but in this work, the results are evaluated based on the combination of the training, validation, and testing.

The ANN model was known for predicting a larger network of dataset that yielded higher accuracy with good training performance but could lead to overfitting the model [21]. Studies by M.S. Zaghloul et al. [21] used the ANN model in predicting complex aerobic granulation in the SBR process. The model was developed using the modular method, in which the first sub-model will predict the biomass characteristics from the predicted biomass characteristics, and it will be used to predict the second sub-model to predict the effluent quality. The network used in this study is a feed-forward neural network (FFNN) with single hidden layer. It consists of large dataset with 2886 samples. The performance shows a compromising result with 99.98% of effluent COD.

Another study from the same author, M.S. Zaghloul et al. [22], reported the comparison between adaptive neuro-fuzzy inference systems (ANFIS) with SVR in predicting AGS. The ANFIS model was a hybrid with the nodes of FFNN in order to handle the fuzzy parameters. The model presented in this work is the continuation work from [21]. From the results, SVR model showed a clear advantage towards the ANFIS model. The ANFIS model struggled with the high level of noise in the data and due to large fuzzy rule bases required the reduction of the number of inputs. This work proved that SVR also shows a great ability to predict a larger dataset, but it needed to optimize a proper setting on the kernel hyperparameters.

### 3.4. Prediction Model of Optimized SVR

The selection of C and γ is crucial in SVR, as it was provided as an input and influenced the way the hyperplane divided based on training data. The optimal parameters are needed, as it will make the best regression model. In practice, the parameters are usually decided by grid search method. The parameters are varied with a fixed step size through a wide range of values, and the performance of every combination is assessed using accuracy performance. The searching parameters can take a very long time, depending on the size of the given dataset, which is time-consuming. Due to computational complexity, this grid search method is only suitable for the adjustment of very few parameters. Hence, we proposed a particle swarm optimization (PSO) and genetic algorithm (GA) for hyperparameters selection.

The development of SVR is further improved by using an optimization method such as particle swarm optimization (PSO) and genetic algorithm (GA). The selection of hyperparameter using conventional grid search method has several disadvantages. To obtain optimal kernel parameters, an improved algorithm is needed. Hence, PSO and GA algorithms are adopted in this SVR model. 

To implement SVR–PSO and SVM–GA approach, the fitness function is designed. PSO and GA contain multiple parameters that need to be set, such as C1, C2, population size, crossover rate, generation size, and maximum iteration. In order to emphasize each optimization, the same setting was implemented in both GA and PSO optimization. The value of inertia weight $w_{\min}$ and $w_{max}$, including C1, C2, are obtained from successful PSO/GA implementation in [56,57,58]. The parameters setting for both PSO and GA are shown in Table 5.

The position of each particle within the swarm in PSO is viewed as a vector encoding the value of SVR parameters, which are regulation and kernel parameters (C and γ). Meanwhile, in GA, the survival principle of the fittest member in the population tries to re-train genetic information from generation to another generation. Through the crossover, a new generation will be created in the current population. The fitness function in these SVR–PSO and SVR–GA stopped when the stopping condition of R^2^ was achieved. Thus, particles with high regression accuracy will produce the best value of fitness. 

Figure 9 shows the comparison between the conventional method of grid search and optimized PSO and GA.

From the plotted graph, the result shows a significant increment in the performance for optimized PSO and GA methods. The performance for SBR30 °C, SVR–PSO, and SVR–GA obtained 98.04% and 98.03%, respectively. It shows an increment of 10% for predicting the effluent COD concentration of 30 °C. The improvement was also shown by SBR40 °C and SBR50 °C for SVR–PSO with 92% and 94%, respectively. The increment is about 5% from the conventional method of grid search. Hence, optimization methods such as PSO and GA are a useful tool in determining optimal solution in predicting complex behaviour of AGS. The capability of each optimization helps to increase the performance of the limited data of this study. The overall performance of the optimization model is tabulated as shown in Table 6.

### 3.5. Model Validation and Evaluation

The validation of the SVR–PSO, SVR–GA, and SVR–Grid Search models are certified based on the validity of the training and testing dataset. The evaluation and validation of the data will be presented by plotting the cross plots of the developed models. The residual plots help to verify the stochastic nature of intrinsic errors that occur in the model. The comparisons of the results are based on three important criteria, which are the lowest value of RSS, the highest value of R2, and the lowest value of RMSE.

As shown in Figure 10, the SVR–PSO and SVR–GA models for SBR30 °C are in line with the regression line of the actual data of COD concentration, with a correlation value of 98.04% and 98.02%, respectively. However, for SVR–Grid Search, the correlation obtained is slightly lower, with 92.24%. This is because the optimizations PSO and GA allowed for more freedom in searching the parameter in hyperdimensional space that resulted in finding global optimum among many local optima. Corresponding to the R^2^ results, the RMSE values obtained are 0.0486, 0.0487, and 0.1021, respectively, which are closer to value of zero. Both models are good and acceptable, as the RSS yields 0.0189, 0.0189, and 0.0834, respectively. Note that RSS value that is closer to zero indicates that the model has a smaller random error component and will be useful for predicting a model. The performance can be further seen in other reactors as well: SBR40 °C with correlation values of 92.54%, 91.82%, and 89.84%; SBR50 °C with correlation values of 94.59%, 94.11%, and 90.05%. The overall validation results for SBR30 °C, SBR40 °C, and SBR50 °C are presented in Table 7. The *p*-value (P1 and P2) in Table 7 indicates the probability of the coefficient (with 95% confidence bounds), and adjusted R^2^ is the best indicator for fit when the value is close to 1.

## 4. Conclusions

Two models were developed in this work, which are SVR and FFNN, to predict the concentration of COD in aerobic granular sludge at three different temperatures (30 °C, 40 °C, and 50 °C). The model was compared for prediction performance using conventional and optimization method, especially when it involved complex systems such as aerobic granulation sludge with a limited dataset. Therefore, some conclusions that can be drawn from this investigation paper are as follows: Based on limited training dataset, the testing evaluation of FFNN performance shows that the accuracy of FFNN is unstable. This is due to the changing number of neurons in the hidden layers with the random combination of weights and threshold values. By contrast, SVR performance shows a stable prediction accuracy with every combination of C and γ selection.

As a comparison, in limited experimental dataset for both FFNN and SVR, SVR is able to find the global solution better than FFNN during the training prediction and obtained excellent testing evaluation with good generalization capability. Meanwhile, FFNN faced a problem such that it easily converged to local minima and faced the overfitting in testing evaluation results.

For the proposed optimized methods of SVR–PSO and SVR–GA, both methods obtained much lower computational times as compared with the SVR–Grid Search. As presented in this works, the CPU time consumed by SVR–Grid Search had much higher ranges from 0.75 to 3.25 h depending on the iteration’s combinations of the parameters. In this works, the pair of C and γ in $\{2^{-15},\;2^{-14},\ldots ,\;2^{14},2^{15}\}$, the grid search produced 31 × 31 = 961 combinations. Meanwhile, the CPU time consumed by SVR–PSO ranged from 0.21 to 1.30 h. Depending on the population/generation size given, which is 100 in this work, the computational time consumed by SVR–PSO is superior to SVR–GA in convergence speed. The CPU time consumed by SVR–GA ranges from 0.32 to 1.58 h.

Analysis of SVR model shows that SVR is an excellent tool for limited dataset as compared to FFNN model. The conventional method of tuning parameters of SVR were then improved by using optimization methods which are PSO and GA. The limited dataset of only 21 has been successfully developed to predict the effluent COD concentration. The results of optimize SVR shows an improvement accuracy of about 10% compared to grid search method.

In many types of applications, SVR can guarantee a higher accuracy for long-term prediction cases compared to the other computational approaches. Fundamentally, three major advantages of SVR can be highlighted throughout this work. First, there are only three types of kernel parameters needed to determine, which are kernel function and cost of the penalty C and gamma γ values. Secondly, the SVR model is highly unique, able to solve global problems and able to provide an optimal solution for solving a linear constrained quadratic problem. Thirdly, SVR provides good generalization performance due to the implementation of structural risk minimization (SRM) principle. However, there is some limitation of this study. A better result of AGS could be better achieved if the dataset used is more than 21 datasets for better division between training and testing datasets. Insufficient sample size could also reduce the performance results for statistical measurement. To further improve the objective function evaluation, the Gravitational Search Algorithm (GSA) method can be considered by a researcher in solving optimization, as it was reported to give better convergent, especially in the WWTP problem. Computational time of each optimization should be recorded to observe the time taken to obtain the highest performance.

## Figures and Tables

**Figure 1 membranes-11-00554-f001:**
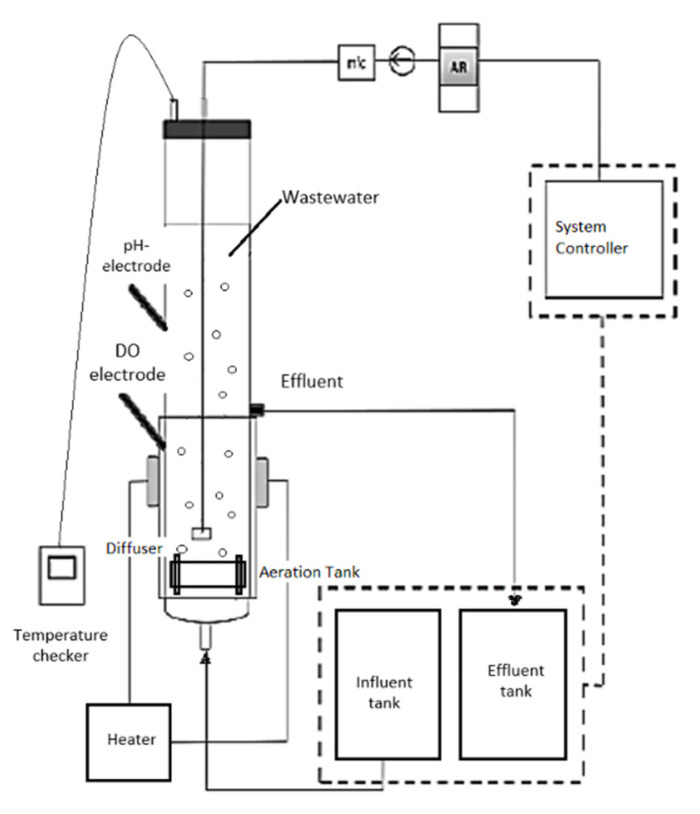
The process of wastewater treatment.

**Figure 2 membranes-11-00554-f002:**
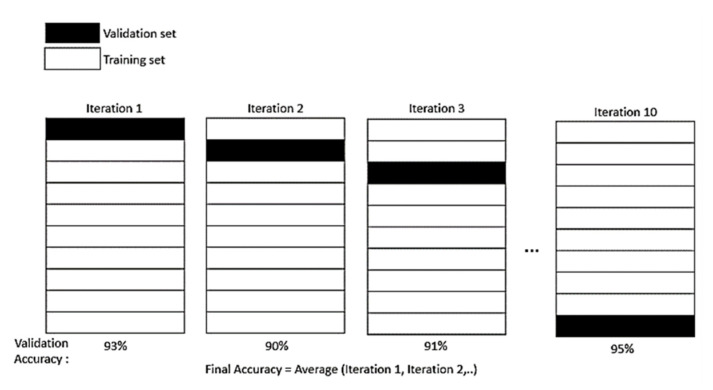
*K*−1 fold CV method.

**Figure 3 membranes-11-00554-f003:**
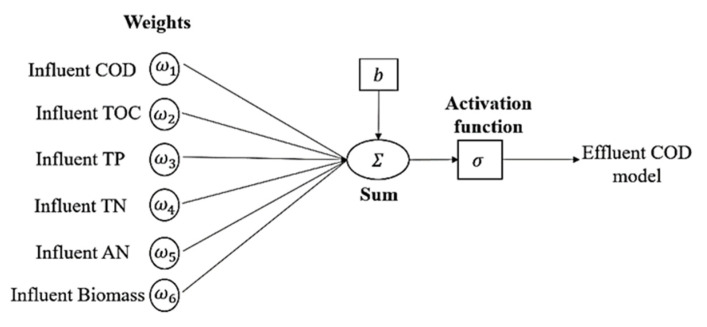
Single neuron of neural network with multiple inputs used.

**Figure 4 membranes-11-00554-f004:**
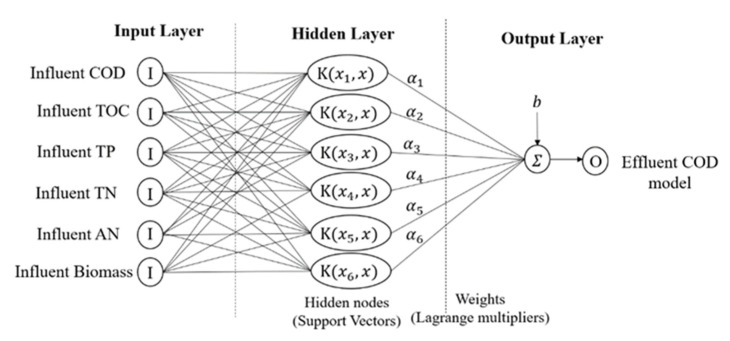
SVR model structure.

**Figure 5 membranes-11-00554-f005:**
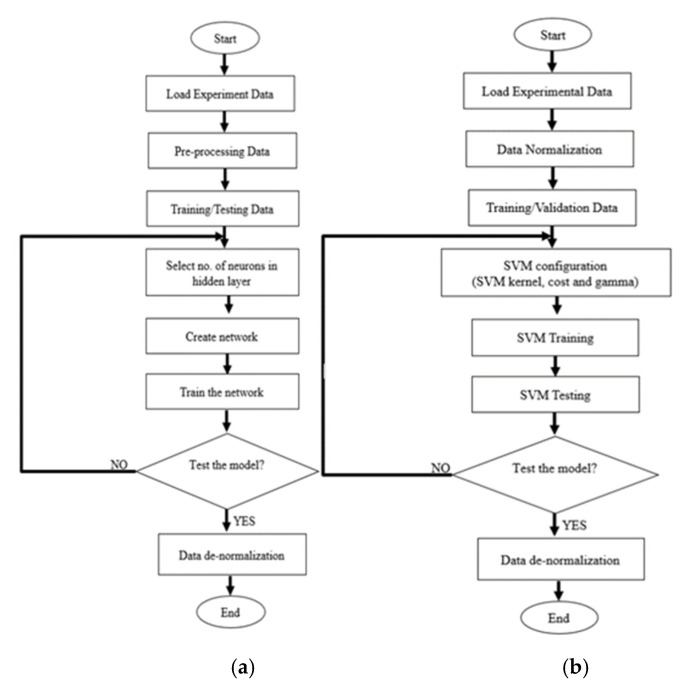
Flow chart (**a**) ANN and (**b**) SVR prediction model.

**Figure 6 membranes-11-00554-f006:**
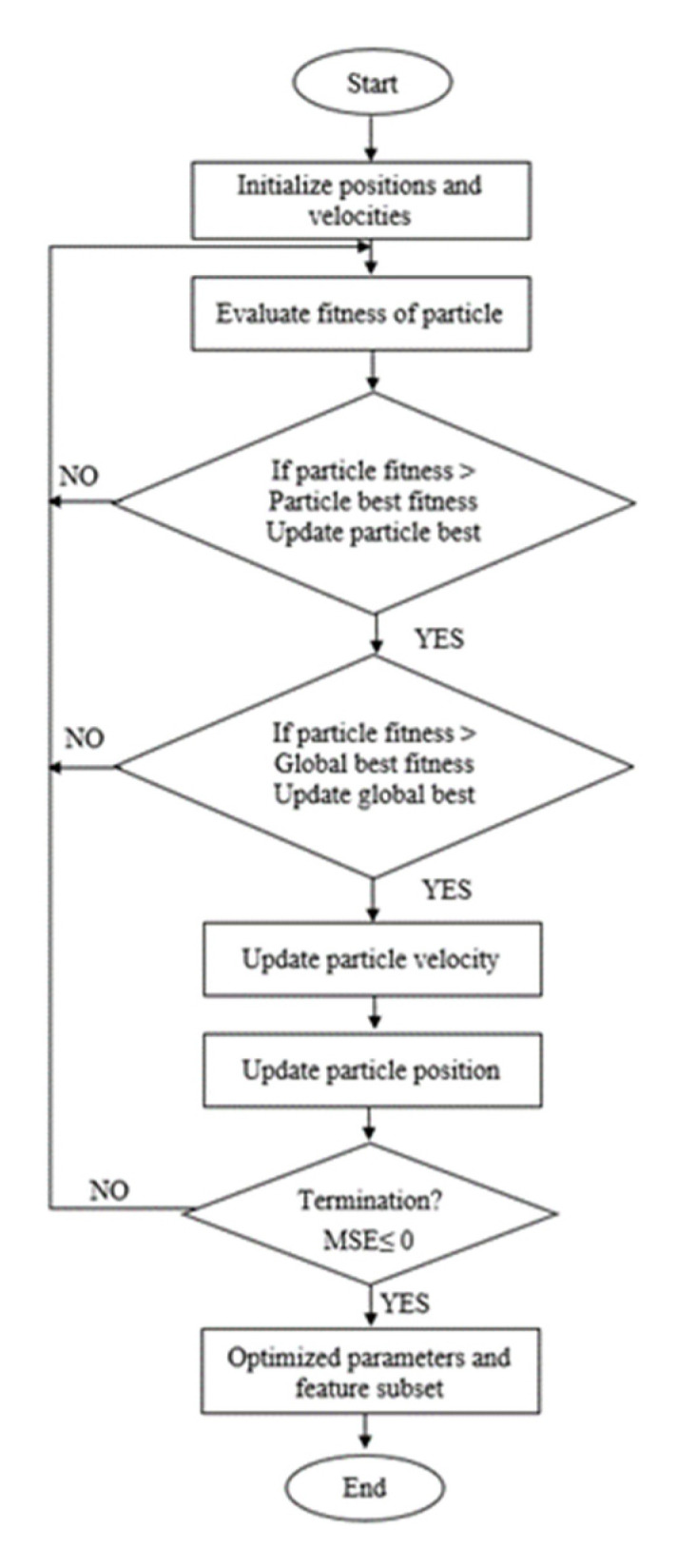
Flow of PSO optimization.

**Figure 7 membranes-11-00554-f007:**
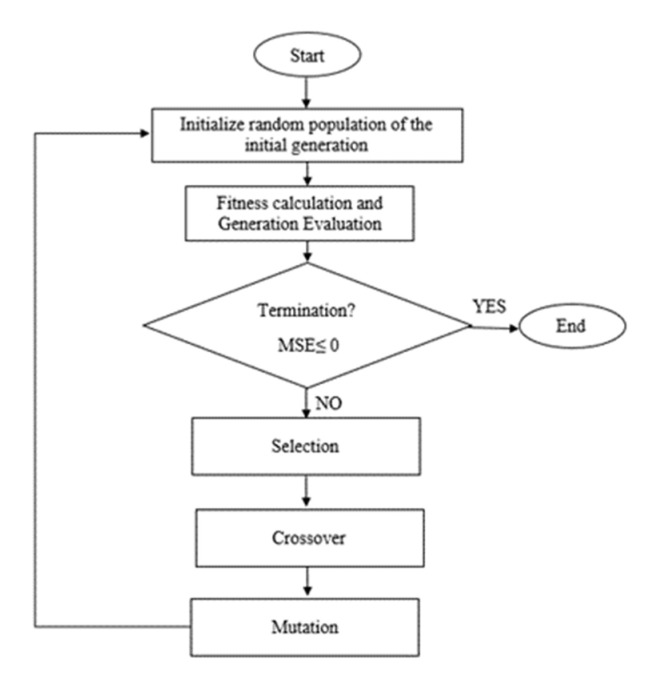
Flow of GA optimization.

**Figure 8 membranes-11-00554-f008:**
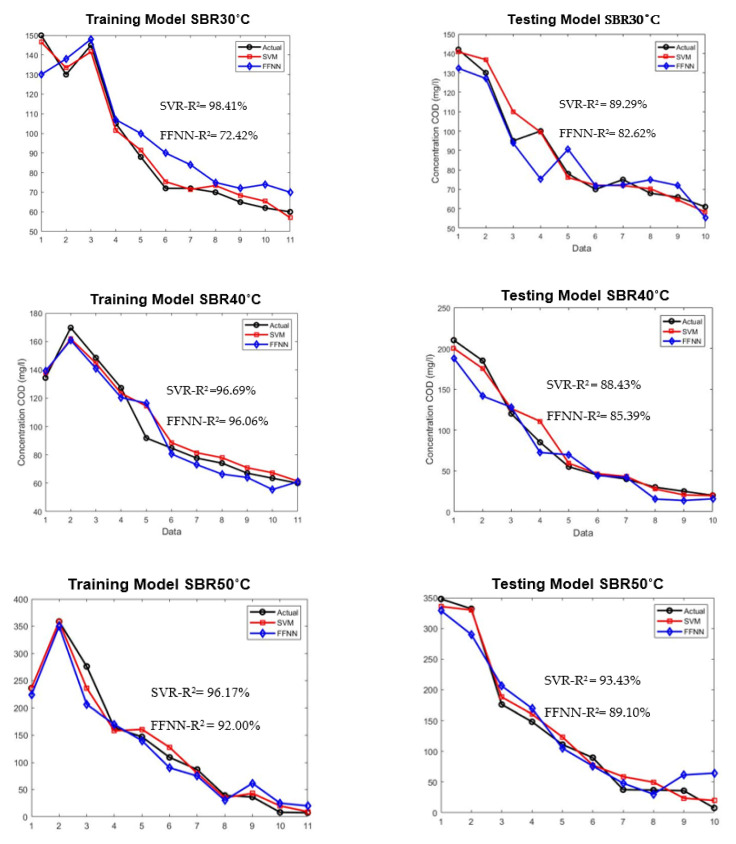
Model development for training and testing of SVR and FFNN for SBR30 °C, 40 °C, and 50 °C.

**Figure 9 membranes-11-00554-f009:**
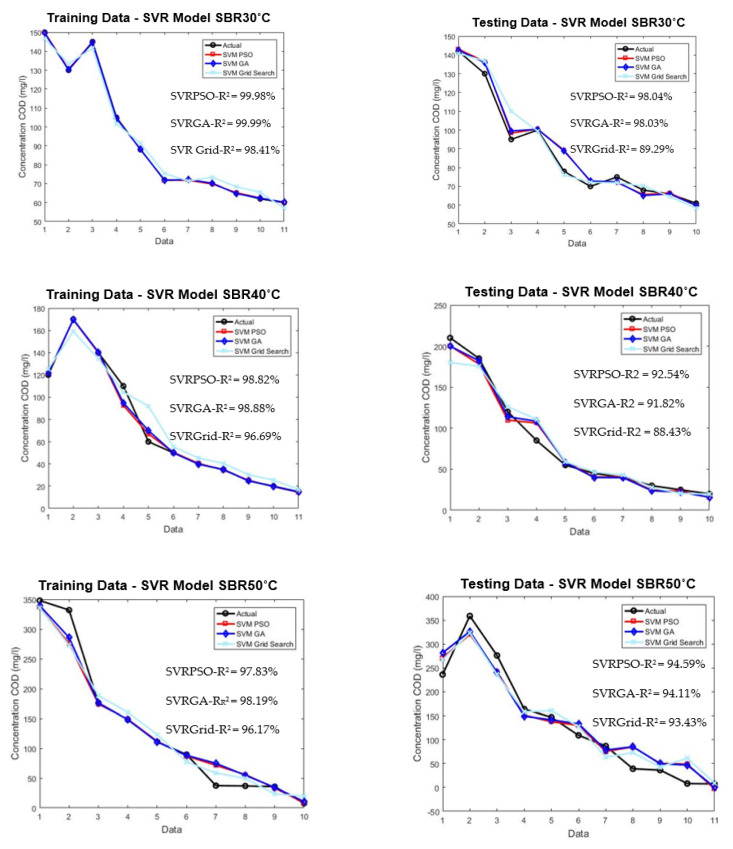
Plotted optimized SVR model for training and testing for SBR30 °C, SBR40 °C, and SBR50 °C.

**Figure 10 membranes-11-00554-f010:**
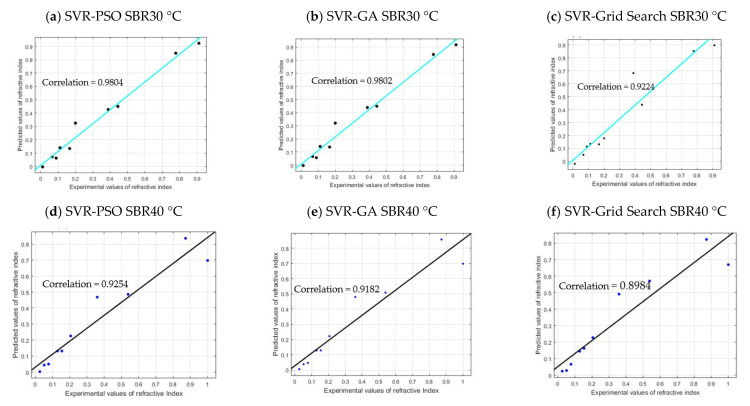
Cross plot validation of each model.

**Table 1 membranes-11-00554-t001:** Influent and effluent parameters statistical analysis.

	Parameters	Units	Max	Min	Mean	Std. Dev.	Median
**Influent**	COD	mg/L	410	390	399.52	8.65	400
	TOC	mg/L	247.50	245.70	246.73	0.64	246.90
	TP	mg/L	21	19	20	0.71	20
	TN	mg/L	54	51	52.67	1.20	53
	AN	mg/L	56	54	55	0.77	55
	Biomass	g/L	10.60	2.70	6.28	2.44	6.50
**Effluent**	COD	mg/L	198	60	92.43	36.63	75
	TOC	mg/L	120.50	11.56	37.34	31.72	22.9
	TP	mg/L	11.80	6	7.88	1.88	6.90
	TN	mg/L	16.50	7.60	10.84	2.88	10
	AN	mg/L	17.10	0.60	5.49	4.35	5.10
	Biomass	g/L	6.50	1.40	4.50	1.90	5.30

**Table 2 membranes-11-00554-t002:** Performance of training SVR models with hyperparameters selection for SBR30 °C, SBR40 °C, and SBR50 °C (the best models are in bold).

Model	Iteration	Gamma, γ	Cost, C	Epsilon, ε	RMSE SBR30 °C	RMSE SBR40 °C	RMSE SBR50 °C
SVR	1	3.05 × 10^−5^	3.05 × 10^−5^	0.01	0.1439	0.1155	0.1238
	49	2	6.10 × 10^−5^	0.01	0.1439	0.1155	0.1238
	97	0.000122	0.000244	0.01	0.1439	0.1155	0.1238
	145	16	0.000488	0.01	0.1438	0.1155	0.1238
	193	0.000977	0.001953	0.01	0.1438	0.1155	0.1238
	241	128	0.003906	0.01	0.1438	0.1155	0.1239
	289	0.007813	0.015625	0.01	0.1424	0.1142	0.1215
	337	1024	0.03125	0.01	0.1436	0.1149	0.1248
	385	0.0625	0.125	0.01	0.0980	0.0882	0.0916
	433	8192	0.25	0.01	0.1353	0.1139	0.1321
	481	0.5	1	0.01	0.0447	0.0356	0.0624
	529	3.05 × 10^−^^5^	4	0.01	0.1423	0.1141	0.1211
	**577**	**4**	**8**	**0.01**	0.0887	0.0170	**0.0128**
	**540**	**0.0625**	**4**	**0.01**	0.0887	**0.0123**	0.0150
	541	0.125	4	0.01	0.0887	0.0868	0.1106
	625	0.000244	32	0.01	0.0879	0.0852	0.0946
	673	32	64	0.01	0.1068	0.1017	0.1193
	**696**	**0.125**	**128**	**0.01**	**0.0208**	0.0153	0.0151
	721	0.001953	256	0.01	0.0290	0.0595	0.0194
	769	256	512	0.01	0.1226	0.1042	0.1195
	817	0.015625	2048	0.01	0.0358	0.1429	0.0621
	865	2048	4096	0.01	0.1240	0.1042	0.1195
	913	0.125	16,384	0.01	0.0208	0.0153	0.0151
	961	32768	32,768	0.01	0.1240	0.1042	0.1195

**Table 3 membranes-11-00554-t003:** Performance of FFNN models for training and testing with different hidden neurons (the best model is in bold).

SBR	No of Hidden Neurons	Training			Testing		
		R^2^ (%)	MSE	RMSE	R^2^ (%)	MSE	RMSE
**FFNN** **30** **°C**	1	78.76	0.0280	0.1672	71.50	0.0249	0.1578
	2	87.84	0.0160	0.1265	77.71	0.0195	0.1395
	4	82.15	0.0064	0.1533	77.46	0.0226	0.1403
	6	81.48	0.0244	0.1561	63.89	0.0315	0.1776
	8	93.63	0.0084	0.0084	75.64	0.0213	0.0213
	**10**	**72.42**	**0.0118**	**0.1084**	**82.62**	**0.0027**	**0.0520**
	12	87.34	0.0167	0.1292	79.61	0.0178	0.1334
	14	79.64	0.0082	0.0264	74.64	0.0221	0.1488
	16	82.15	0.0099	0.0994	71.28	0.0218	0.1476
**FFNN** **40** **°C**	1	68.98	0.0216	0.1471	71.86	0.0312	0.1765
	2	87.84	0.0085	0.0921	81.80	0.0201	0.1419
	4	75.80	0.0169	0.1299	80.19	0.0131	0.1143
	6	89.27	0.0075	0.0865	84.78	0.0169	0.1298
	8	77.86	0.0154	0.1242	80.73	0.0103	0.1140
	10	83.63	0.0114	0.1068	84.53	0.0138	0.1175
	12	81.36	0.0130	0.1140	84.58	0.0171	0.1307
	**14**	**96.06**	**0.0027**	**0.0524**	**85.39**	**0.0162**	**0.1672**
	16	98.5	0.001	0.0324	78.15	0.0242	0.1555
**FFNN 50** **°C**	1	65.01	0.0349	0.1868	60.73	0.0426	0.2064
	2	91.28	0.0087	0.0933	84.15	0.0172	0.1312
	4	95.83	0.0042	0.0645	88.21	0.0128	0.01131
	**6**	**92.00**	**0.0087**	**0.2949**	**89.10**	**0.0109**	**0.1044**
	8	93.23	0.0067	0.0822	84.2	0.0171	0.1309
	10	95.39	0.0046	0.0678	8654	0.0146	0.1207
	12	90.97	0.0090	0.0949	86.79	0.0143	0.1197
	14	80.60	0.0193	0.1391	85.43	0.0158	0.1258
	16	81.51	0.0184	0.1358	78.60	0.0221	0.1488

**Table 4 membranes-11-00554-t004:** Comparison works between this work with previous work.

Author	This Work (2021)					
Collected Data	COD Model: 21					
Model	SVR			FFNN		
	R^2^ (%)		MSE	R^2^ (%)	MSE	
**SBR30 °C**	89.29		0.0095	82.62	0.0207	
**SBR40 °C**	88.43		0.0131	85.39	0.1672	
**SBR50 °C**	93.43		0.0065	89.10	0.0109	
**Author**	**M.S. Zaghloul et al. (2018)**					
**Collected Data**	**COD Model: 2686**					
**Model**	**SVR**			**FFNN**		
	**R^2^ (%)**	**MSE**		**R^2^ (%)**		**MSE**
**COD (mg/L)**	99.98	0.2701		98.85		0.1211
**Author**	**M.S. Zaghloul et al. (2020)**					
**Collected Data**	**COD Model: 2920**					
**Model**	**ANFIS**			**SVR**		
	**R^2^ (%)**	**MSE**		**R^2^ (%)**	**MSE**	
**COD (mg/L)**	98.50	0.348		99.99	0.035	
**Author**	**H. Gong et. al. (2018)**					
**Collected Data**	**COD Model: 205 and 136**					
**Model**	**ANN**					
	**R^2^ (%)**			**MSE**		
**COD (mg/L)**	90.00			2.399		

**Table 5 membranes-11-00554-t005:** Parameters setting for PSO and GA.

C1	2.05
C2	2.05
Population/generation size	20
Crossover rate	0.9
$w_{\min}$	0.4
$w_{\max}$	0.9
Maximum iteration	100

**Table 6 membranes-11-00554-t006:** Performance comparison between optimize SVR models at different temperature.

SBR	Model	Training		Testing	
		R² (%)	MSE	R² (%)	MSE
**30** **°C**	**SVR–PSO**	99.98	1.1898 × 10^−5^	98.04	0.0023
	**SVR–GA**	99.99	7.8769 × 10^−^	98.03	0.0024
	**SVR–Grid Search**	98.41	0.0012	89.29	0.0095
**40** **°C**	**SVR–PSO**	98.82	8.5298 × 10^−4^	92.54	0.0109
	**SVR–GA**	98.88	7.9146 × 10^−4^	91.82	0.0109
	**SVR–Grid Search**	96.69	0.00032	88.43	0.01312
**50** **°C**	**SVR–PSO**	97.83	0.0040	94.59	0.0064
	**SVR–GA**	98.19	0.0032	94.11	0.0067
	**SVR–Grid Search**	96.17	0.0041	93.43	0.0065

**Table 7 membranes-11-00554-t007:** Goodness of fit for each model.

Goodness of Fit SBR30 °C	SVR–PSO		SVR–GA		SVR–Grid Search	
	Training	Testing	Training	Testing	Training	Testing
**RSS**	0.0012	0.0189	9.492 × 10^−5^	0.0189	0.0105	0.0834
**R2**	0.9999	0.9804	0.9999	0.9802	0.9921	0.9224
**Adjusted R2**	0.9999	0.9780	0.9999	0.9778	0.9912	0.9127
**RMSE**	0.0037	0.0486	0.0032	0.0487	0.0341	0.1021
**P1**	0.9979	1.043	0.9938	1.0380	0.9535	1.065
**P2**	0.0007	0.0078	0.0017	0.0085	0.0236	0.0088
**Goodness of Fit** **SBR40 °C**	**SVR–PSO**		**SVR–GA**		**SVR–Grid Search**	
	**Training**	**Testing**	**Training**	**Testing**	**Training**	**Testing**
**RSS**	0.0086	0.0588	0.0084	0.0679	0.0239	0.0781
**R2**	0.9882	0.9254	0.9888	0.9182	0.9629	0.8984
**Adjusted R2**	0.9896	0.9161	0.9875	0.9079	0.9588	0.8857
**RMSE**	0.0310	0.0857	0.0305	0.0921	0.0516	0.0988
**P1**	0.9755	0.8116	0.9813	0.8297	0.0046	0.7899
**P2**	0.0024	0.0304	0.0034	0.0275	−0.0197	0.0512
**Goodness of Fit** **SBR50 °C**	**SVR–PSO**		**SVR–GA**		**SVR–Grid Search**	
	**Training**	**Testing**	**Training**	**Testing**	**Training**	**Testing**
**RSS**	0.0178	0.0456	0.0153	0.0515	0.0241	0.0446
**R2**	0.9783	0.9459	0.9819	0.9411	0.9715	0.9465
**Adjusted R2**	0.9755	0.9399	0.9796	0.9345	0.9680	0.9405
**RMSE**	0.0471	0.0712	0.0438	0.0756	0.0549	0.0704
**P1**	0.8593	0.8529	0.8764	0.8665	0.8722	0.8481
**P2**	0.0445	0.06321	0.0440	0.0644	0.0416	0.0684

## Data Availability

Not applicable.

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
