# Peer review of "Support Vector Regression Modelling of an Aerobic Granular Sludge in Sequential Batch Reactor"

_membranes, 2021, doi:10.3390/membranes11080554_

Round 1

Reviewer 1 Report

The authors report an experimental study of chemical oxygen demand in aerobic granular sludge processes using sequential batch reactors. Support vector regression (SVR) models are developed and compared to feed forward neural network (FFNN) models. SVR parameters were optimized using three different techniques: 1) grid search, 2) particle swarm, and 3) genetic algorithm. The results suggest both models provide good predictions and the SVR models are slightly better than the FFNN models for the limited datasets used.

I recommend that the authors address the following comments before publication:

  1. General comment. The manuscript would benefit from extensive editing for grammar and spelling.

  2. Page 1. The first sentences of Section 1.1 (Background) are directions for writing the introduction. This text should be deleted.

  3. Page 4. Please define GA and PSO before their use in the text.

  4. Page 5. Please indicate the aerator and diffuser components in Figure 1. Additionally, Figure 1 suggests the outer wall of the double wall reactor extends only halfway to the top of the reactor. Is this correct?

  5. Page 5. The authors state that the following parameters are considered: 1) total nitrogen (TN), 2) total phosphorus (TP), 3) ammonia nitrogen (AN), 4) total organic carbon (TOC), 5) mixed liquor suspended solids (MLSS), and 6) chemical oxygen demand (COD). However, the authors do not indicate how the measurements are made. Additionally, it is not clear if the authors are reporting new measurements or simply using measurements taken from the literature (reference 34). Please clarify if the experimental results are new and the instrumentation used to obtain the data.

  6. Page 5. The authors should report the raw experimental data as supplemental material.

  7. Page 6. The authors report many of the measurements in Table 1 to four significant figures. Are the results accurate to four significant figures? Please report using the correct number.

  8. Page 7. In Equation 2, the subscript on v is missing.

  9. Page 8. In Equation 4, parentheses are missing around the difference between the sum and theta. Is this correct?

  10. Page 8. In Equation 5, the vector theta is omitted. Is this correct?

  11. Page 8. The discussion of SVR should be rewritten to improve clarity. The discussion provided does not convey the theory behind or implementation of SVR well.

  12. Page 8. The authors appear to state “i” corresponds to the number of data points but “n” is the number of data points. Please clarify.

  13. Page 13. Please define y bar in Equation (16).

  14. Page 14. Please revise the text that begins with “Various” in the first paragraph. This word does not appear to be part of a sentence.

  15. Page 14. The text states epsilon is set equal to one. However, Table 2 suggest the value used is 0.1. Please clarify.

  16. Page 16. The authors claim to demonstrate SVR is superior to FFNN for complex, limited datasets. What is the error in the experimental data used to fit the models? Are the differences in the predictions within the experimental error? How can the authors generalize their findings to other situations?

  17. Page 17. Please show training and testing data for both SVR and FFNN in Figure 8.

  18. Page 18. The authors should describe the parameters used in the GA and PSO searches more thoroughly. Additionally, please comment on the computational time required for both methods.

  19. Page 21. Please define the adjusted R2, P1, and P2 values reported in Table 6.

  20. Page 21. The authors should address how experimental error affects their conclusions about which model best fits the experimental data. Are the differences between the model predictions comparable to the experimental error? What is the statistical significance of the differences?

  21. Page 21. The authors compare fits for three different algorithms used to determine SVR parameter values. However, they do not report the computational time used to perform the calculations. It is not clear that a better fit with GA or PSO is simply the result of increased computational time – not the result of a more efficient search. Please provide additional discussion.

Author Response

Please see the attachment. Thank you for your comment. The manuscript has been revised accordingly.

The authors report an experimental study of chemical oxygen demand in aerobic granular sludge processes using sequential batch reactors. Support vector regression (SVR) models are developed and compared to feed-forward neural network (FFNN) models. SVR parameters were optimized using three different techniques: 1) grid search, 2) particle swarm, and 3) genetic algorithm. The results suggest both models provide good predictions and the SVR models are slightly better than the FFNN models for the limited datasets used.

I recommend that the authors address the following comments before publication:

1. General comment. The manuscript would benefit from extensive editing for grammar and spelling.

                                                                        The authors take note of the comment and improvement will yet be made.

2. Page 1. The first sentences of Section 1.1 (Background) are directions for writing the introduction. This text should be deleted.

Page 1. The correction has been made.

The authors do appreciate the thorough review and the manuscript has been revised accordingly.

3. Page 4. Please define GA and PSO before their use in the text.

Page 4. Line 152-153. GA and PSO are defined accordingly.

4. Page 5. Please indicate the aerator and diffuser components in Figure 1. Additionally, Figure 1 suggests the outer wall of the double-wall reactor extends only halfway to the top of the reactor. Is this correct?

Page 5. The aerator and diffuser have been added to the schematic diagram in Figure 1.

5. Page 5. The authors state that the following parameters are considered: 1) total nitrogen (TN), 2) total phosphorus (TP), 3) ammonia nitrogen (AN), 4) total organic carbon (TOC), 5) mixed liquor suspended solids (MLSS), and 6) chemical oxygen demand (COD). However, the authors do not indicate how the measurements are made. Additionally, it is not clear if the authors are reporting new measurements or simply using measurements taken from the literature (reference 34). Please clarify if the experimental results are new and the instrumentation used to obtain the data.

Page 5. Line 216.

Section 2.2

A reference [33] was added in the text. The experiment was done by [33].

6. Page 5. The authors should report the raw experimental data as supplemental material.

Page 5. The raw experimental normalize data (for 30oC) was provided in the supplemental material. However, only sample data was provided as the data is collaboration data. Hence, it was advised to keep private and confidential.

7. Page 6. The authors report many of the measurements in Table 1 to four significant figures. Are the results accurate to four significant figures? Please report using the correct number.

Page 6. The results in Table 1 have been updated using the correct number.

8. Page 7. In Equation 2, the subscript on v is missing.

Page 7. The correction has been made in Eq. (2).

9. Page 8. In Equation 4, parentheses are missing around the difference between the sum and theta. Is this correct?

Page 8. The correction has been made in Eq. (4).

10. Page 8. In Equation 5, the vector theta is omitted. Is this correct?

Page 8. The correction has been made in Eq. (5).

11.   Page 8. The discussion of SVR should be rewritten to improve clarity. The discussion provided does not convey the theory behind the implementation of SVR well.

Page 8. The discussion of SVR in Section 2.5 reflected the implementation of SVR in this work. Additional discussion on kernel functions and parameters C has been added on lines number 319-322 and 335.

12. Page 8. The authors appear to state “i” corresponds to the number of data points, but “n” is the number of data points. Please clarify.

Page 8. Line numbers 308. The correction has been made.

13. Page 13. Please define y bar in Equation (16).

Page 13. Line 424.

Y bar is the mean of the predicted data.

14. Page 14. Please revise the text that begins with “Various” in the first paragraph. This word does not appear to be part of a sentence.

Page 14.

The authors do appreciate the thorough review and the manuscript has been revised accordingly.

15. Page 14. The text states epsilon is set equal to one. However, Table 2 suggests the value used is 0.01. Please clarify.

Page 14. The correction has been made.

The authors do appreciate the thorough review and the manuscript has been revised accordingly.

16. Page 16. The authors claim to demonstrate SVR is superior to FFNN for complex, limited datasets. What is the error in the experimental data used to fit the models? Are the differences in the predictions within the experimental error? How can the authors generalize their findings to other situations?

As stated in the manuscript in Sec. 2.9, the results are made comparison based on residual sum of square due to error (RSS), R2 and MSE.

RSS value that closer to 0 indicates that the model has a smaller random error component and that the fit will be more useful for prediction.

Meanwhile, R2 is defined as the ratio of the sum of squares of the regression (RSS) and the total sum of squares (SST). Hence, examine the sum of squares due to error (SSE) and the adjusted R-square statistics to help determine the best fit. All the statistical values of the goodness of fit are tabulated as shown in Table 7.

17. Page 17. Please show training and testing data for both SVR and FFNN in Figure 8.

Page 17. Both comparison graph of training and testing data for SVR and FFNN has been added in Figure 8.

18. Page 18. The authors should describe the parameters used in the GA and PSO searches more thoroughly. Additionally, please comment on the computational time required for both methods.

Page 18. The parameters used in PSO and GA have been added as described in Table 5. The computational time has been added in Section 4.

19. Page 21. Please define the adjusted R2, P1, and P2 values reported in Table 6.

Page 20. Line 631. The Adjusted R2, P1 and P2 have been defined in the paragraph.

20. Page 21. The authors should address how experimental error affects their conclusions about which model best fits the experimental data. Are the differences between the model predictions comparable to the experimental error? What is the statistical significance of the differences?

As stated in the manuscript in Section 2.9, the results are made comparison based on residual sum of square due to error (RSS), R2 and MSE.

RSS value that closer to 0 indicates that the model has a smaller random error component and that the fit will be more useful for prediction.

Meanwhile, R2 is defined as the ratio of the sum of squares of the regression (RSS) and the total sum of squares (SST). Hence, examine the sum of squares due to error (SSE) and the adjusted R-square statistics to help determine the best fit. All the statistical values of the goodness of fit are tabulated as shown in Table 7.

21. Page 21. The authors compare fits for three different algorithms used to determine SVR parameter values. However, they do not report the computational time used to perform the calculations. It is not clear that a better fit with GA or PSO is simply the result of increased computational time – not the result of a more efficient search. Please provide additional discussion.

Page 22. Line 651-660. The computational time used for choosing parameters in SVR has been reported in Section 4. (Conclusion).

Reviewer 2 Report

The article has scientific and practical value. Results obtained by the authors showed the potential of SVR for simulating the complex aerobic granulation process. Lines 27-35 are unnecessary, do not belong to the article. The layout of the article needs to be corrected.

Author Response

Please see the attachment. Thank you for your comment. The authors do appreciate the thorough review and the manuscript has been revised accordingly.

Reviewer 3 Report

The manuscript ‘Support vector regression modelling of an aerobic granular sludge in sequential batch reactor’ is a very interesting paper and has important practical aspects. The scientific approach adopted in the study is rigorous and the paper is also well-organized. Therefore, I believe that manuscript should be published in the Membranes in present form.

Author Response

(The authors gave the same response as above.)

Round 2

Reviewer 1 Report

The authors have not adequately addressed several issues.

  1. Figure 1. Figure 1 indicates the heating mantle extends half way up the AGS column. Please explain why half the column is not heated.
  2. Table 1. The authors were asked to use the appropriate number of significant figures in Table 1. The revised manuscript reports more significant figures instead of less as recommended in the review. This must be justified. For example, the mean COD is reported to 5 significant figures. This implies the results are accurate to 5 significant figures (two decimal places). Do they believe their measurements are accurate to two decimal places given instrument error and sample variability? Similar comments apply for the other parameters.

  3. Discussion of experimental error. The authors do not adequately address experimental error. While the correct statistical measures of goodness of fit are reported, the authors do not provide an indication of experimental error (a combination of measurement error and sample-to-sample variability) and its magnitude relative to the differences in model predictions. For example:

    a. Indicate the experimental error with error bars for points in Figures 8-10.
    b. Please provide estimates of the sample-to-sample variability and the measurement error.
    c. The authors appear to weight all data points equally when minimizing the error. Please justify this approximation over the range of operating and measurement conditions used.
  4. Conclusion. A portion of the conclusion is numbered. Why?

Round 3

Reviewer 1 Report

The authors have tried to address reviewer concerns.